# The Effect of Time Synchronization Error in LAN-Based Digital Substation

**DOI:** 10.3390/s19092044

**Published:** 2019-05-01

**Authors:** Kyou Jung Son, Tae Gyu Chang, Sang-Hee Kang

**Affiliations:** 1School of Electrical and Electronics Engineering, Chung-Ang University, Seoul 06974, Korea; skj9865@naver.com; 2Department of Electrical Engineering, Myongji University, Yongin 17058, Korea; shkang@mju.ac.kr

**Keywords:** IEEE 1588 precision time protocol, time synchronization error, intelligent electronic device, IEC 61850-based substation

## Abstract

In this paper, the effect of time synchronization error on protection algorithms are studied for the usage of the LAN-based collaborative protection. In order to derive the effect of time synchronization, this paper proposes a substation model which is constructed with IEEE 1588 Precision Time Protocol (PTP) supported intelligent electronic devices. The proposed model is used as an example of a target platform to study the effect of time synchronization error with two typical substation protection algorithms, i.e., current differential-based substation protection and distance protection algorithms. From the analyzed and the simulated results, it was well observed that time synchronization error is a significant error-causing factor for both protection algorithms, resulting in erroneous detection of faults and erroneous estimation of fault distances, respectively. The results of research performed in this paper are expected to provide a good guide for constructing the future LAN-based digital power substation with precise time synchronization.

## 1. Introduction

Future LAN-based digital substation is expected to provide collaborative protections with the utilization of wider range of time synchronized data acquired from the LAN-based distributed sensing environment. Therefore, the provision of precise time synchronization is a very important issue for the stable and reliable operation of the substation [1,2,3,4,5]. Especially, as suggested in [6,7], advanced optical sensing technology increases the feasibility of the high throughput and the reliability of future LAN-based digital substations.

However, the implementation of the precise time synchronization is generally considered as a difficult task since it requires optimization of synchronizing parameters in connection with protective algorithmic details. Moreover, the proper handling of the fault causing factors in the LAN-based environment, such as failure of time synchronizing device, packet loss in heavy network, external security attack including GPS disturbance, etc., must be considered in the design and operation of the precise time synchronization.

The standard and profiles, IEEE 1588 Precision Time Protocol (PTP) [8], IEEE c37.238 [9] and IEC 61850-9-3 [10] provide specifications of precise time synchronization and communication protocol for the future LAN-based digital substation. Among previous researches on time synchronization applications in power system applications, PTP performance analysis in heavy networks [1,2], PTP performance on the IEC 61850 SV process bus [11,12,13] and PTP performance degradation in clock failure [14] are examples of results which show the effect of fault causing factors in a LAN-based substation. Previous research on PMU (Phasor Measurement Unit) performance degradation in GPS failure [15,16], PTP performance degradation in time desynchronization attacks [17], and a detection and mitigation model for PTP delay attacks [18], dealt with the reliability issues of time synchronization under external disturbances and attacks. The results achieved through these previous researches are useful in the sense that they provide the performance of time synchronization under various failure conditions.

For the optimized realization of time synchronization, domain specific protection features must be jointly considered to provide collaborative and advanced protection functions for future LAN-based digital substations. The joint consideration reflecting domain specific protection details is a difficult task to achieve since it requires comprehensive studies on substation protections in relation with time synchronization. The difficulty causes the lack of joint consideration in previous research and leads to the restrictive focuses on the performance of time synchronization itself.

In this paper, two typical protection algorithms, i.e., a current differential-based substation protection algorithm [19] and a distance protection algorithm, are selected as a target for studying the effect of time synchronization error in protection algorithms. This is considered as a significant contribution for advanced time synchronization in the sense that it can provide an example of joint consideration reflecting protection algorithm specific features of a LAN-based smart substation. The two protection algorithms were selected since they are widely used in power system applications and show the typical usage of distributed measurement data showing relatively high-sensitivity to time synchronization error. The effects of time synchronization error in the two protection algorithms were analyzed. The results of the analysis can provide important references in designing reliable time synchronization reflecting the performance degradation of collaborative protections in the LAN-based substation. The analysis is also an exemplary illustration of the usage of distributed and time synchronized measurement data for collaborative and advanced protection of a substation.

A substation model which was constructed with IEEE 1588 PTP supported intelligent electronic devices (IEDs), is proposed in order to study the effect of time synchronization error. Computer simulations and analytic derivation were also performed for the validation of the effect of time synchronization error in the distributed sensing environment.

The current differential-based substation protection algorithm [19] was simulated with simplified 154 kV substation with 14 measurement points. Maloperation ratio of the protection algorithm was simulated with three cases of single line-to-ground faults having different fault points for illustrating the effect of time synchronization error. From the simulated results, it is confirmed that time synchronization error is a significant factor in causing the occurrences of erroneous fault-detection.

The effect of time synchronization error on the distance protection algorithm was analyzed and simulated with two source-based power system for the line-to-line fault case. From the analytic derivation of formula for per unit distance error, it is shown that time synchronization error appears inside the argument of tangent function in both real and imaginary parts of a relay estimating impedance. The validation of the analytical results is shown from the results of simulations. The simulation results also showed a significant amount of a per unit distance error was caused by time synchronization error.

## 2. Distributed Sensing in Power System Environment

In this section, a time synchronizing IED-based substation model is proposed. The proposed substation model is considered as a basic and simplified structure to realize a collaborative protection in the future IEC 61850-based smart substation. The substation model will be used as an example of a target platform to study the effect of time synchronization error in typical substation protection algorithms in Section 3 and Section 4.

The basic structure of the proposed time synchronizing substation model consists of collaborative IEDs connected to the wired/wireless LAN-based network as shown in Figure 1. In the proposed time synchronizing substation model, the collaborative IED is a core constructing element which integrates the functions of collaborative protection and time synchronization. The integrated design and operation of the algorithms in each IED platform is required for the adaptive configuration of time synchronization, where, in general, the PTP network topology, PTP control parameters, accuracy class of a clock, etc., must be configured adaptively according to the required precision level of each protection element. The adaptive configuration of time synchronization is needed to reflect time synchronization error detection and recovery functions in connection with collaborative protection algorithms of IEDs.

For the purpose of adaptive configuration of time synchronization, the two network-based protocols, i.e., IEC 61850 communication networks and systems for power systems [20] and IEEE 1588 precise time synchronization [8], should be implemented together with the protection and time synchronization algorithms in the same IED platform.

The time synchronization structure of the proposed substation model is designed with a master-slave based time synchronization protocol which is supported with hardware timestamping structure [8]. This structure provides sub-microsecond synchronization accuracy among the collaborative IEDs regardless of the wired or wireless network [8,21,22].

Reliability of digital substation cannot be achieved without providing the functions of time synchronization error detection and recovery. Master clock failure detection is a typical example of the collaborative operation of IEDs. The master clock failure detection is based on sharing of time correction data among the collaborative IEDs. The shared data are processed for the application of thresholding logic. As an example of detecting the master’s abnormal PTP time offset, the time windowed PTP time offset sequences are compared and thresholded with the data collected from distributed slave IEDs. As suggested in the IEC 61850-9-3 communication protocol, the TLV (type, length, value) message is a good choice to utilize for the sharing of PTP data [8,10].

In the proposed substation model, a recovery management IED is suggested to share the TLV messages for the adaptive configuration of time synchronization during the PTP master failure. The core PTP parameters including backup run-time, clock accuracy class, synchronization period, etc., must be set adaptively reflecting the time synchronization error effects to the performance of protection algorithms. Therefore, the failure detection and recovery must be designed to realize flexible and adaptive profiles of substation protection reflecting the domain specific protections and algorithms.

However, the optimized failure detection and recovery is very difficult to achieve because it requires a comprehensive study on the performance of substation protection and the effects of time synchronization error. In Section 3 and Section 4, narrowing down to the two typical protection algorithms, i.e., current differential-based substation protection algorithm and distance protection algorithm, the effect of time synchronization error were analyzed.

## 3. Application of Precise Time Synchronization for Collaborative Protection of Power System

In this section, based on the substation model proposed in Section 2, two typical examples of protection algorithms in power substation were studied to show the effect of time synchronization error. A current differential-based collaborative protection system was designed with the application of the IEEE 1588 PTP to show the occurrences of erroneous fault-detection caused by time synchronization error. Distance relay’s fault locating error caused by the impreciseness of time synchronization was also studied for the analytic derivation of formula for per unit distance error.

### 3.1. Current Differential-Based Collaborative Protection

Functional blocks required for the design of current differential-based collaborative protection system are shown in Figure 2. The collaborative protection consists of a main fault detection block and auxiliary blocks for correction and alignment of measurement data collected from distributed locations. Adaptive operation of the protection algorithm can be achieved by configuring protection-related parameters which reflects the varying status of a substation.

The current differential-based main fault detection block detects a fault in the substation. The protection algorithm applies thresholding logic to the differential current value which is obtained from Kirchhoff’s Current Law (KCL)-based summation of the measurement data. Normalization of the measurement data should be performed before obtaining the differential current value since the measurement data includes primary and secondary current values of a transformer.

The correction and compensation block produces the uncorrupted measurement data. Initialization process corrects the parameters used by the sensing units to obtain unbiased V/I (voltage and current) information. Compensation of nonlinearity was performed to prevent the distortion of measurement data caused by the saturation effect of an instrumental transformer.

The PTP based measurement alignment block provides the synchronized measurement data from the distributed sensing units. A master-slave based time synchronization protocol guarantees precisely synchronized operation of the sensing units. The asynchronized measurement timing of each sensing unit is compensated by applying signal processing technique such as extra/interpolation of the measurement values.

By analyzing the status of substation including load variation and transient of a fault, the protection algorithm operates adaptively. According to the status of substation, the parameters used by the protection algorithm and PTP structure are adaptively configured.

In Section 4.1, as a target simulation of current differential-based collaborative protection, substation-area backup protection (SBP) algorithm [19] was used to show the performance degradation caused by time synchronization error.

### 3.2. Analysis of the Effect of Time Synchronization Error in Digital Distance Protection Algorithm

In this subsection, the effect of time synchronization error in measurement data on a digital distance protection algorithm for line-to-line fault case was analyzed. Time synchronization error was applied to B-phase of V/I signal under AB fault case. The operation of digital distance protection was based on the measurement and evaluation of the short-circuit impedance by using the phasors of discrete voltage and current signal [21]. For each of fault types, different formulas should be adopted when calculating the fault impedance. Table 1 indicates calculation formula for all the fault types [23,24].

Under AB fault situation, the currents of A and B-phases, $I_{A}$ and $I_{B}$, have the same magnitude but opposite sign. It is assumed that time synchronization error is applied to B-phase V/I measurement IED and the error appears as a time delay index D. Then, the voltage and current signals can be expressed as Equations (1)–(4):(1)$$V_{A}\left(n\right)=V_{m}\cos\left(2\pi fnT_{s}\right)$$
(2)$$V_{B}\left(n\right)=V_{m}\cos\left(2\pi f\left(n-D\right)T_{s}-\frac{2}{3}\pi \right)$$
(3)$$I_{A}\left(n\right)=I_{m}\cos\left(2\pi fnT_{s}+\theta_{1}\right)$$
(4)$$I_{B}\left(n\right)=-I_{m}\cos\left(2\pi f\left(n-D\right)T_{s}+\theta_{1}\right)$$
where $V_{m}$ and $I_{m}$ denote the magnitude of voltage and current respectively, f denotes the frequency of power signal $\theta_{1}$ denotes the phase of current with respect to $V_{A}$, $T_{s}$ denotes sampling time and n denotes time index. D denotes the time delay index caused by time synchronization error.

The phasor of the fundamental frequency of $V_{A}$ is expressed as Equation (5):(5)$$H_{V_{A}}=V_{m}X,\;X=\underset{n=\langle N\rangle }{\sum }cos\left(2\pi fnT_{s}\right)e^{-i\frac{2\pi k}{N}n}.$$

The phasors of the signals can be expressed as Equations (6)–(8), as well:(6)$$H_{V_{B}}=V_{m}Xe^{-i\frac{2}{3}\pi }e^{-i2\pi fDT_{s}}$$
(7)$$H_{I_{A}}=I_{m}Xe^{i\theta_{1}}$$
(8)$$H_{I_{B}}=-I_{m}Xe^{i\theta_{1}}e^{-i2\pi fDT_{s}}$$

From Table 1, the relay estimating impedance $Z_{AB}$ for AB fault case can be calculated as Equation (9):(9)$$Z_{AB}=\frac{V_{m}}{I_{m}e^{i\theta_{1}}}\cdot \frac{\left(1-e^{-i\frac{2}{3}\pi }e^{-iC}\right)}{\left(1+e^{-iC}\right)},\;C=2\pi fDT_{s}.$$

The second multiplication term of Equation (9) can be simplified by using Euler’s formula and trigonometric function:(10)$$\frac{1-e^{-i\frac{2}{3}\pi }e^{-iC}}{1+e^{-iC}}=\;\frac{1}{4}\left(3+\sqrt{3}\tan\left(C\right)+j\left(\sqrt{3}+\tan\left(C\right)\right)\right)\;∴\;Z_{AB}=\frac{V_{m}}{4I_{m}}\left(3+\sqrt{3}\tan\left(C\right)+j\left(\sqrt{3}+\tan\left(C\right)\right)\right)e^{-i\theta_{1}}.$$

The real and imaginary parts of Equation (10) can be divided as Equation (11):(11)$$Z_{AB}=\frac{V_{m}}{4I_{m}}\left(\left(\sqrt{3}+\tan\left(C\right)\right)S_{R}+j\left(\sqrt{3}+\tan\left(C\right)\right)S_{I}\right),$$
(12)$$S_{R}=\sqrt{3}\cos(\theta_{1})+\sin\left(\theta_{1}\right),$$
(13)$$S_{I}=\cos\left(\theta_{1}\right)-\sqrt{3}\sin\left(\theta_{1}\right).$$

Equation (11) shows time synchronization error in a distance relaying algorithm under AB fault case appears inside the argument of tangent function in both real and imaginary parts of a relay estimating impedance $Z_{AB}$. Considering the feature of tangent function, this means the relay estimating impedance $Z_{AB}$ can be very far from its true impedance depending on time synchronization error. This results in malfunction of the distance relaying algorithm, so the stable operation of the power system will be failed.

In this subsection, the effect of time synchronization error between phases in the distance protection algorithm was analyzed. In the future distributed sensing-based substation, traditional fixed measurement format will be changed into the freed and unconstrained measurement format for applying various protecting and monitoring approaches. The analytical result obtained from in the subsection is expected to provide basic information for constructing the future distributed sensing-based substation.

## 4. Simulations and Results

Reflection of domain specific protection details for the designing of an advanced time synchronizing system in power substation is generally a difficult task since it requires comprehensive exploitation of inter-relationships among the many and various protection components in the substation. The effect of time synchronization error on protection algorithms in power system has not been much studied in previous research. However, the effect of time synchronization error in substation protection studied in this paper is compared with representative studies by Bi [15] and Zhang [16]. As summarized in Table 2, the major difference of the precision range was found to come from the time synchronization platform difference between LAN-based PTP and dedicated line-based synchronization.

In the following, the performance degradation caused by time synchronization error of two typical protection algorithms in power system are simulated. Current differential-based substation-area backup protection (SBP) algorithm [19] is simulated for a 154-kV substation model with 14 measurement points. For the simulation of the distance protection algorithm, two source-based power system was modeled to obtain fault data.

### 4.1. Performance Degradation of Collaborative Protection Algorithm Caused by Time Synchronization Error

Substation-area Backup Protection (SBP) [19] is an algorithm based on the current differential principle for locating the faulted component in the IEC 61850-based substation and its adjacent IEC 61850-based substations. As a substitute for the traditional backup protection, SBP provides three functions, i.e., redundant primary protection, optimized backup protection, and integrated automatic control. The details of these functions are illustrated in [19] (pp. 2–3).

Simplified diagram of a 154-kV substation is shown in Figure 3. Faults 1–3 denote fault components used for simulations of the effect of time synchronization error on the SBP algorithm. L1 and L2 denote transmission lines. A1 and A2 denote an example of layered zones to localize a fault component. To locate a fault component, SBP algorithm divides the substation into many zones. From large zone to small zone, SBP detects where a fault has occurred by using Equation (14):(14)$$I_{diff}>I_{thres}$$
(15)$$I_{diff}=\left|\overset{N_{x}}{\underset{i=1}{\sum }}{\dot{I}}_{i}\right|$$
where $I_{diff}$ is differential current and $I_{thres}$ is threshold value which depends on the level of restraint current $I_{R}=\frac{1}{2}\left(\sum_{i=1}^{N_{x}}\left|\dot{I_{i}}\right|\right)$. $N_{x}$ is the total number of branches in the *x*-zone and ${\dot{I}}_{i}$ is a current phasor.

Suppose that a transmission line fault occurs at fault 1 and SBP tries to find the fault component. SBP will start up due to zone A1 being in operating (fault) state. If SBP firstly tries to remove L2 from zone A1, the remaining zone (A2+L1) is still in operating state. Furthermore, after L1 is removed, remaining zone A2 immediately turns to restraining (fault released) state. Thus, it is determined that the fault lies at L1 [19]. With further detailed divisions of the zones, fault 2 and 3 can be found by applying similar method.

Time synchronization error in SBP causes unaligned phasors which come from asynchronized sampled values among measuring points. This affects calculating $I_{diff}$ which is obtained from (15). To understand how SBP operates under time asynchronized situation, fault simulations were performed by using MATLAB/Simulink program. Various parameters used in the modeled power system and simulation experiments are shown in Table 3.

In the simulation, it is assumed that time synchronization error occurs at the only one measuring point among 14 points in each simulation. The range of synchronization error is from 50 to 450 μs and for each error case 14 simulations are performed to consider all the case of time synchronization error in single measuring point.

The performance of SBP algorithm is represented as the percentage of maloperation which was defined as 2 fault conditions, i.e., SBP fails to detect a fault or to find the location of a fault. The result of performance of SBP algorithm for the case of Fault 2 is shown in Figure 4. As time synchronization error increases, the maloperation ratio of SBP increases. When time synchronization error is 250 μs which is close to sampling time, SBP shows about 7% of maloperation ratio under load 100% situation. This means asynchronized measurement timing among sensing units can cause 7% error of detecting Fault 2. Precise time synchronization among sensing units should be preceded to apply current differential-based protection.

Higher load usage is more vulnerable to time synchronization error than lower load usage as shown in Figure 4. This is because higher load usage causes larger fault current on a transmission line than lower load usage. For the same amount of synchronization error, larger fault current results in a more erroneous calculation of $I_{diff}$. Table 4 shows the average of maloperation ratio of SBP for Fault 1–3. As expected, SBP is more vulnerable to time synchronization error when a load usage is high.

### 4.2. Validation of the Analytical Result of the Effect of Time Synchronization Error in Digital Distance Protection Relay

To validate the analytical result of the effect of time synchronization error in an algorithm for digital distance protection, two source-based power systems have been modeled which are shown in Figure 5. Various parameters used in the power system and simulation are shown in Table 5. Fault simulations were performed by using MATLAB/Simulink program and compared with the analytical result. Time synchronization error is realized by delaying B-phase voltage and current data. The range of time synchronization error was −15–+15 sample.

Comparison of the distance relay estimating resistance and the resistance obtained from the analytically derived formula is shown in Figure 6a. The validation of the derived formula is confirmed considering the small difference between the resistances. The shape of graph follows the shape of tangent function with respect to the delay parameter as expected. As delay grows apart from zero, the values of the resistances go farther from the true value with increasing rate of change.

Comparison of the distance relay estimating reactance and the reactance obtained from analytically derived formula is shown in Figure 6b. Likewise, the small difference between the resistances validates the derived formula. The shape of graph follows the shape of tangent function with respect to the delay parameter.

To investigate the effect of time synchronization error in the operation of the distance protection algorithm, AB-fault simulations with varying time synchronization error on B-phase signal were performed for the mho type distance protection relay. Figure 7 shows a fault impedance locus for a fault at 70 km without time synchronization error. The distance protection algorithm detects the occurrence of zone 1 fault without time synchronization error.

Figure 8a–c show fault impedance traces for faults at 70 km with 2, 3 and 5 samples of time synchronization error on B-phase signal. As time synchronization error increases, the distance protection algorithm locates the fault far away from the original fault distance. The estimated fault distances for varying time synchronization error are shown in Table 6 where it is observed that the per unit distance error ranges from 0.06% to 92.72% when the time synchronization error varies from 1 sample to 10 samples.

## 5. Conclusions

In this paper, the effects of time synchronization error are analytically derived and simulated to show the occurrences of erroneous fault detection. The time synchronizing IED-based substation model is proposed to study time synchronization error effects with the two typical substation protection algorithms. A significant amount of maloperation ratio, which reaches to near 10%, is shown as a result of time synchronization error bound to 250 μs for the case of the current differential-based protection algorithm. A significant amount of a per unit distance error is also observed which ranges from 0.06% to 92.72% when the time synchronization error varies from 1 sample to 10 samples for the case of the distance protection algorithm. The importance of accurate and fault tolerate time synchronization, which are considered as essential for the realization of collaborative protection for future LAN-based digital power substations, is well illustrated through the results of this research. Also, the results of the analysis can be considered as important references in designing the reliable time synchronization reflecting the performance degradation of collaborative protections.

## Figures and Tables

**Figure 1 sensors-19-02044-f001:**
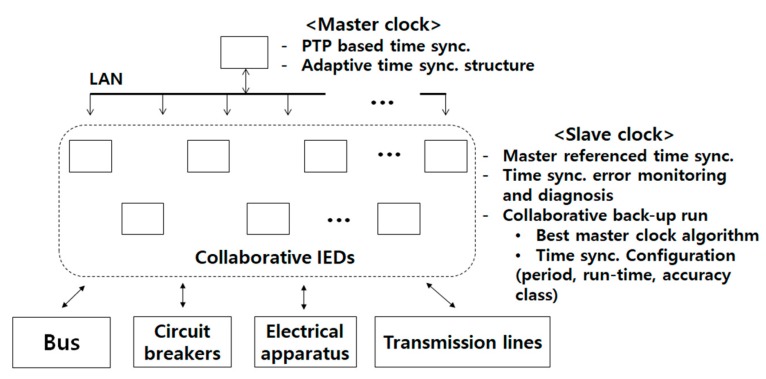
Time synchronizing Intelligent Electronic Device (IED)-based substation model proposed for IEC 61850-based smart substation.

**Figure 2 sensors-19-02044-f002:**
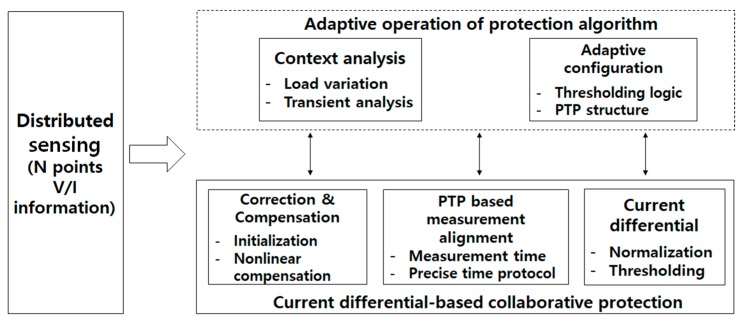
Functional blocks of current differential-based collaborative protection system based on the application of the IEEE 1588 Precision Time Protocol (PTP).

**Figure 3 sensors-19-02044-f003:**
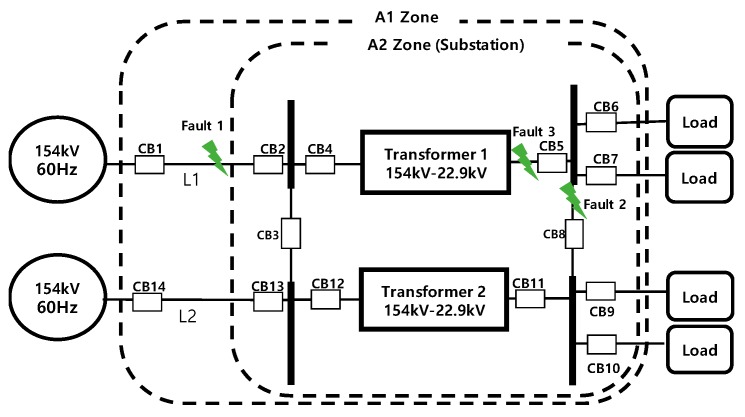
An example of simplified diagram of a 154-kV substation.

**Figure 4 sensors-19-02044-f004:**
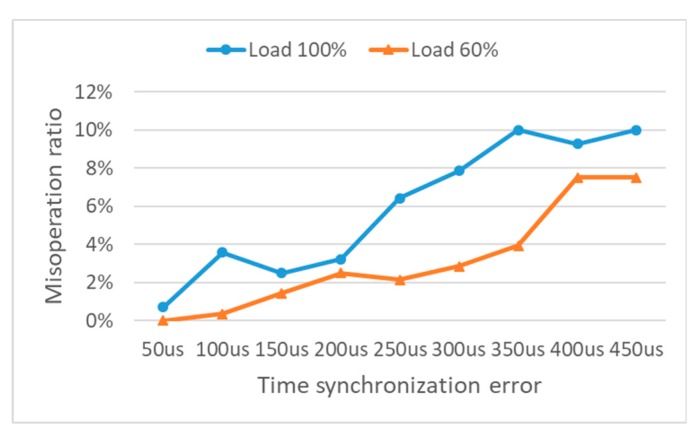
The result of performance of Substation-area Backup Protection (SBP) algorithm for Fault 2 case.

**Figure 5 sensors-19-02044-f005:**
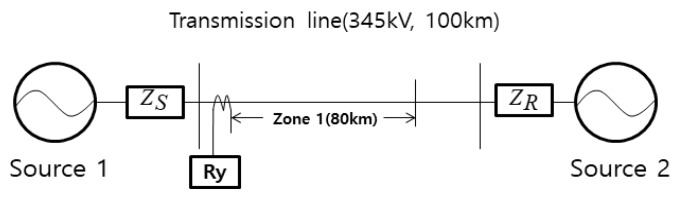
Two sources-based power system for simulations of the effect of time synchronization on distance protection algorithm.

**Figure 6 sensors-19-02044-f006:**
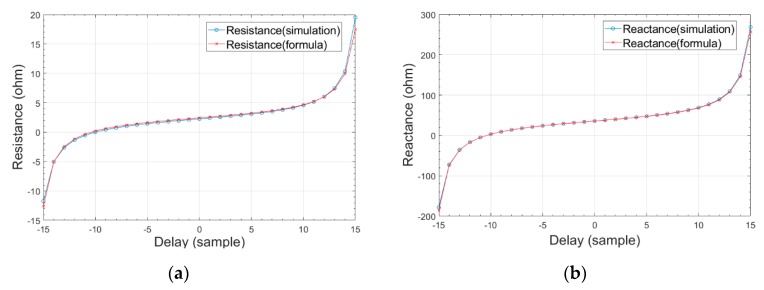
Comparison of distance relay estimated impedance and analytically obtained impedance under AB fault case. (**a**) resistance, (**b**) reactance.

**Figure 7 sensors-19-02044-f007:**
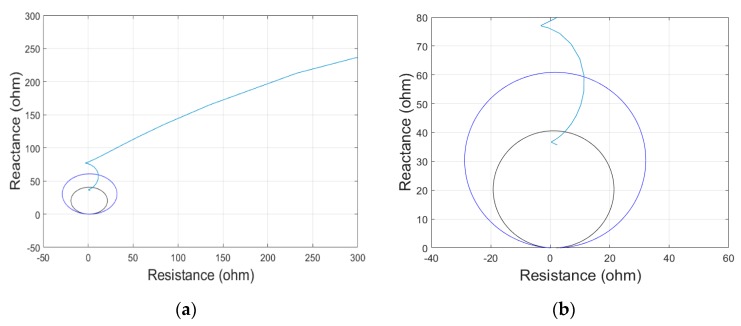
Fault impedance locus for fault at 70 km without time synchronization error. (**a**) locus of distance relay estimating impedance, (**b**) zoom in view of distance relay protection zone.

**Figure 8 sensors-19-02044-f008:**
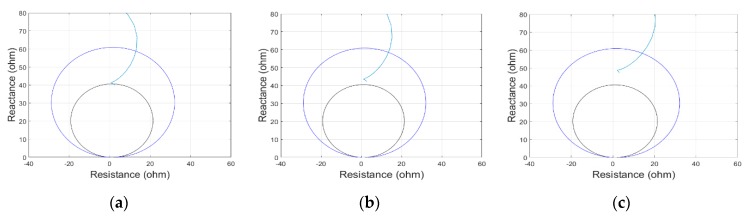
Fault impedance loci for a fault at 70km with time synchronization error on the B-phase signal. (**a**) two samples, (**b**) three samples, (**c**) five samples.

**Table 1 sensors-19-02044-t001:** Fault impedance calculation formula on difference faults.

Fault Type	Formula
AB	$\left(V_{A}-V_{B}\right)/\left(I_{A}-I_{B}\right)$
BC	$\left(V_{B}-V_{C}\right)/\left(I_{B}-I_{C}\right)$
CA	$\left(V_{C}-V_{A}\right)/\left(I_{C}-I_{A}\right)$
AG	$V_{A}/\left(I_{A}+3kI_{0}\right)$
BG	$V_{B}/\left(I_{B}+3kI_{0}\right)$
CG	$V_{C}/\left(I_{C}+3kI_{0}\right)$

A, B and C denote each single-phase of three-phase power system, G denotes the ground, AB, BC, and CA denote short circuit fault between phases, AG, BG, and CG denote short circuit fault between a single phase and the ground, V and I are phasors of voltage and current, *k* = $\left(Z_{0}-Z_{1}\right)/Z_{1}$, $Z_{0}$ and $Z_{1}$ are the zero and positive-sequence line impedances respectively, and $I_{0}$ is the zero-sequence current.

**Table 2 sensors-19-02044-t002:** Comparison summary: different approaches of analyzing the effect of time synchronization error in power system.

Comparison Features	Studies		
	This Paper	Bi [15]	Zhang [16]
Protection algorithm used for analysis	Distance protection, Current differential-based protection	Rotor angle measurement, Wide area power system stabilizer	Transmission line fault detection and localization
Target precision range	~7.8 ms	~1 s	~10 s
Platform of synchronization system	IEEE 1588 PTP-based synchronization	Dedicated line-based synchronization	Dedicated line-based synchronization
Purpose of the analysis	Provide a guide for constructing the LAN-based digital substation with precise time synchronization	Analyzing the effect of GPS error on synchrophasor measurement under dynamic state	Validation of the effectiveness of proposed GPS attack model

**Table 3 sensors-19-02044-t003:** Parameters used in computer simulation.

Parameter	Value
Substation capacity	120 MVA
Voltage amplitude	154 kV
Frequency	60 Hz
Sampling frequency	4800 Hz
Fault case	Line-to-ground (AG fault),
Fault location	Fault 1–3
Fault resistance	0.1 Ω
Load consumed power	100%, 60% of substation capacity

**Table 4 sensors-19-02044-t004:** Average of maloperation ratio of SBP for Fault 1–3.

Time Synchronization Error (µs)	Average of Maloperation Ratio (%)	
	Load 100%	Load 60%
50	1.07	0.24
100	3.45	1.55
150	2.98	3.81
200	3.69	3.10
250	6.07	2.98
300	6.90	4.76
350	8.57	5.71
400	7.62	6.67
450	8.21	6.79

**Table 5 sensors-19-02044-t005:** Parameters used in computer simulation.

Parameter	Value
Voltage amplitude	345 kV
Frequency	60 Hz
Sampling frequency	1920 Hz
Fault case	Line-to-line (AB fault)
Fault distance	70 km
Relay Zone 1 setting	80 km
Relay Zone 2 setting	120 km

**Table 6 sensors-19-02044-t006:** Fault distances for varying time synchronization error.

Time Synchronization Error (sample)	Fault Distance (km)
0	70.4768
1	74.7531
2	79.1153
3	83.6568
5	93.7219
7	106.2087
10	135.8202

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
