# Peer review of "The Effect of Time Synchronization Error in LAN-Based Digital Substation"

_sensors, 2019, doi:10.3390/s19092044_

Round 1

Author Response

First of all, we thank the reviewer for the considerations on our submitted manuscript. We have revised the manuscript reflecting all the reviewer's requirements and comments. The changes we have made in the revised manuscript are summarized in the attached word file.

Reviewer 2 Report

Dear Authors, please find here below a list of comments/suggestions that hopefully could be useful for improving your paper.

Please, specify the meaning of the acronym IEDin the abstract, or directly use the form: "intelligent electronic devices".

It is not clear what Authors mean for adpative configuration of time synchronization.

Authors should specify the meaning of some very spesific terms, such as A and B-phase and the effect of the different fault types (AB, BC, CA, AG, BG and CG) in order to make the paper accessible also for non-experts. A very short introduction to this argument might be useful.

Authors should better explained the meaning of the parameter D. This parameter is defined as a delay in raw 181, pag. 5. It is not clear is this parameter represents the synchronization error, or if it effectively represents a delay whose estimation depends also on the synchronization error (see Eq. 4).

Please carefully check the use of English, especially in Section 4. Some sentences and expressions are not correct or very difficult to understand.

Fig. 3 ahould be better explained. In particular, it is not clear the correlation between fault 1-2-3 and the purpose of zone B and C.

In order to understand the meaning of the sentence "when a fault occurs at fault 3", a reference to Fig. 3 should be firstly introduce in the text.

Author Response

(The authors gave the same response as above.)

Round 2

Reviewer 2 Report

In my opinion the paper can be published in its current form.